# Effects of schistosomes on host anti-viral immune response and the acquisition, virulence, and prevention of viral infections: A systematic review

**Brooke W. Bullington**[1]*, **Katherine Klemperer**[2], **Keith Mages**[3], **Andrea Chalem**[1], **Humphrey D. Mazigo**[4], **John Changalucha**[4], **Saidi Kapiga**[4,5], **Peter F. Wright**[6], **Maria M. Yazdanbakhsh**[7], **Jennifer A. Downs**[1]

1 Center for Global Health, Weill Cornell Medicine, New York, NY, United States of America, 2 London School of Hygiene and Tropical Medicine, London, United Kingdom, 3 Samuel J. Wood Library Weill Cornell Medicine, New York, NY, United States of America, 4 Mwanza Intervention Trials Unit, National Institute for Medical Research Mwanza, Tanzania, 5 Department of Infectious Diseases Epidemiology, London School of Hygiene and Tropical Medicine, London, United Kingdom, 6 Department of Pediatrics, Dartmouth Geisel School of Medicine, Hanover, New Hampshire, United States of America, 7 Department of Parasitology, Leiden University Medical Center, Leiden, Netherlands

* bbullington@unc.edu

**Data Availability Statement:** All relevant data are within the manuscript and its Supporting Information files.

## Abstract

Although a growing number of studies suggest interactions between *Schistosoma* parasites and viral infections, the effects of schistosome infections on the host response to viruses have not been evaluated comprehensively. In this systematic review, we investigated how schistosomes impact incidence, virulence, and prevention of viral infections in humans and animals. We also evaluated immune effects of schistosomes in those coinfected with viruses. We screened 4,730 studies and included 103. Schistosomes may increase susceptibility to some viruses, including HIV and Kaposi's sarcoma-associated herpesvirus, and virulence of hepatitis B and C viruses. In contrast, schistosome infection may be protective in chronic HIV, Human T-cell Lymphotropic Virus-Type 1, and respiratory viruses, though further research is needed. Schistosome infections were consistently reported to impair immune responses to hepatitis B and possibly measles vaccines. Understanding the interplay between schistosomes and viruses has ramifications for anti-viral vaccination strategies and global control of viral infections.

## Author summary

Many studies have described the effects of parasitic *Schistosoma* worm infections on the way that humans and animals respond to a variety of viral infections. Our goal was to evaluate, in a systematic manner, how having a schistosome parasitic infection affects a host's susceptibility to viral infections, the clinical disease course of viral infections, and prevention of viral infections by vaccines. We also assessed the effects of schistosome infection on the host immune response to viruses. We screened 4,730 studies for potential relevance

**Funding:** JAD received support from NIH K23 AI110238 (https://www.nih.gov/) and the Doris Duke Charitable Foundation 2017067 (https://www.ddcf.org/). The funders had no role in the study design, analysis, decision to publish, or preparation of the manuscript.

**Competing interests:** The authors have declared that no competing interests exist.

and included 103 of them in this review. Overall, our analysis showed that schistosome infection impairs the host response to many viruses. This includes increasing host susceptibility to HIV and possibly Kaposi's sarcoma-associated herpesvirus, worsening the severity of clinical disease in hepatitis B and C infections, and decreasing immune responses to vaccines for hepatitis B and possibly measles. The studies that we analyzed also suggested that schistosome infection may protect the host against poor clinical outcomes from some viral infections including Human T-cell Lymphotropic Virus-Type 1, respiratory viruses, and chronic HIV. We discuss how these findings might be interpreted, and the additional research needed, in order to improve anti-viral vaccination strategies and control of viral infections globally.

## Introduction

Viral infections caused over 3 million deaths worldwide in 2017 [1,2], and will likely contribute over 1 million more in 2020 due to COVID-19 [3]. Global viral outbreaks including COVID-19, HIV, and influenza illustrate the critical need to understand how and why the same virus may cause mild disease in some hosts, yet trigger severe disease and death in others.

Helminth infections affect over 2 billion people worldwide and disproportionately infect the world's poor [4]. A growing body of literature suggests that helminth infections, particularly schistosomes, affect both host susceptibility to viruses and the severity of viral illness via their impact on the host immune system. While previous reviews have investigated effects of schistosome co-infection on HIV [5–8] or hepatitis viruses [9–11], a more comprehensive analysis of the impact of schistosome infection on viral acquisition, virulence, and prevention has not been undertaken.

Schistosomes infect at least 229 million people globally and are treatable with praziquantel [12]. Parasitic worms lay eggs that damage tissue of the gastrointestinal (*S. mansoni*, *S. japonicum*) and genitourinary (*S. haematobium*) tracts, causing ~1.5 million disability-adjusted life years (DALYs) lost annually [13]. If schistosome infection additionally exacerbates viral susceptibility or virulence, the burden of DALYs lost may be higher. Conversely, if schistosomes benefit the host in some instances, this might hold potential for new therapeutic and prevention strategies against viral infections.

In this review, we systematically examine the effects of schistosome infection on viral infections. We hypothesized that this broader investigation would provide insights to guide future research, with potential to impact the public health effects of viruses. We focused on how schistosome co-infection impacts specific viral infections by examining: host anti-viral immune responses, prevalence and incidence of the viral infection, virulence, and anti-viral vaccination.

## Methods

This systematic review utilized the Preferred Reporting Items for Systematic Reviews and Meta-Analysis (PRISMA) guidelines [14] and was registered prospectively in PROSPERO (CRD42019129533).

### Search strategy

Comprehensive literature searches were developed and performed by a medical librarian [KM]. The initial search was performed March 28, 2019 via OVID MEDLINE ALL (1946-March 27, 2019). The Cochrane Library (Cochrane Database of Systematic Reviews,

Cochrane Central Register of Controlled Trials (CENTRAL), Cochrane Methodology Register, Technology Assessments (HTA)), LILACS (Latin American and Caribbean Health Sciences Literature), and OVID EMBASE (1974-March 28, 2019) were searched on March 29, 2019; Scopus (Elsevier) on April 1, 2019. Follow-up searches were performed on December 9, 2019. Search terms included all subject headings and/or keywords associated with our research question, clustered as:

1. Parasite or proxy for parasitic infection (e.g. Schistosoma, Schistosomiasis);

2. Virus or vaccine-related headings (e.g. Virus, Virology, Vaccine, Vaccination);

3. Immune response or physiologic state indicative of host immune response (e.g. Immuno-modulation, Lymphocyte Activation, Biomarkers, Disease Susceptibility, Cellular Immunity, Coinfection, HIV, T-Lymphocytes).

Boolean operators 'OR' and 'AND' were used as appropriate. Grey literature and bibliographies of included articles were also searched; see full search strategy in the Supplementary Appendix (**S1 Text**). Publication date or article-type restrictions were not imposed.

### Study selection

After excluding duplicates, two investigators [BB, KK] independently screened titles and abstracts using Covidence, a systematic review tool. Discrepancies were resolved independently by a third investigator [JD]. We included studies describing the effects of schistosomes on viral infections, anti-viral immunity, or anti-viral vaccinations and defined virulence as severity of clinical disease in co-infected subjects [15]. This definition of virulence aligned with the definitions used in included manuscripts. Pre-defined exclusion criteria included: populations with and without viruses rather than with and without schistosome infection, focus on non-viruses, not available in English, and study design of case study or case review. We pre-specified that studies must test for active schistosome infection by microscopy or antigen and that we would exclude studies that relied on schistosome antibody testing due to antibodies remaining positive after clearance of infection (see **S2 Text** for full decision rules for data selection and extraction).

Full-text review followed the initial title and abstract screening phase, with data extracted into Microsoft Excel. Following Cochrane Collaboration guidance, included articles were evaluated for credibility, transferability, dependability, and confirmability [16] using Critical Appraisal Skills Programme checklists [17]. Discrepant analyses of study quality were resolved by discussion and author consensus. Due to heterogeneity of study designs, animal and human studies, and different viral infections, a meta-analysis was not possible.

### Results

Titles and abstracts of 4,731 studies were screened, and 313 full-text papers were assessed for inclusion. 14 additional records were identified from the grey literature, expert recommendations, and screening bibliographies. Excluded studies are listed in **S1 Table**. A total of 103 studies were included (**Fig 1** and **S2 Table**).

We identified studies examining effects of schistosome coinfection on 7 categories of viruses: hepatotropic viruses such as Hepatitis B and C (HBV, HCV), Human Immunodeficiency Virus (HIV), Human papillomavirus (HPV), Human T-cell leukemia virus type 1 (HTLV-1), Kaposi's sarcoma-associated herpesvirus (KSHV), respiratory viruses (influenza, murine herpesviruses), and rubeola virus vaccination.

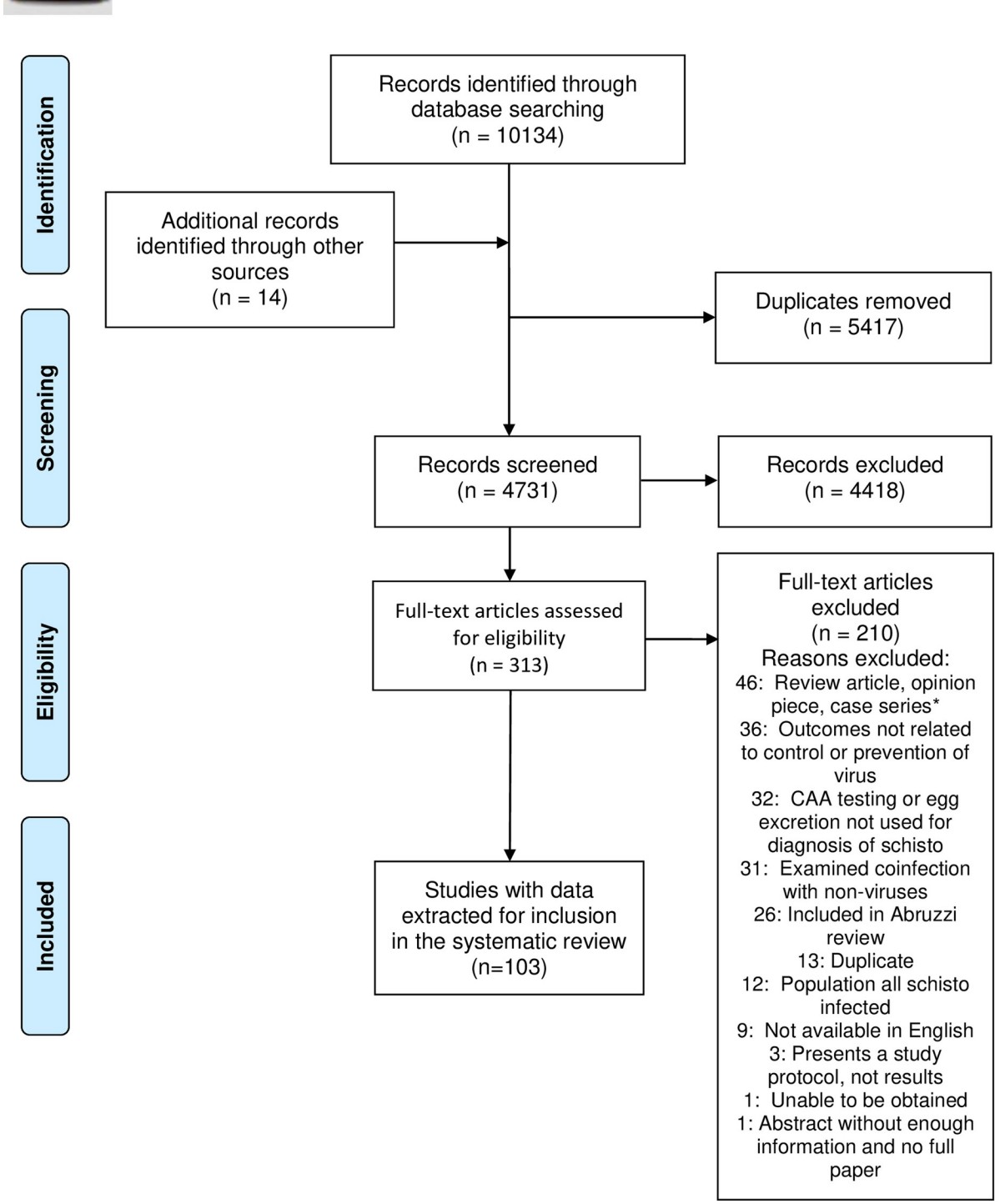

**PRISMA 2009 Flow Diagram**

*From:* Moher D, Liberati A, Tetzlaff J, Altman DG, The PRISMA Group (2009). *P*referred *R*eporting *I*tems for *S*ystematic Reviews and *M*eta-*A*nalyses: The PRISMA Statement. PLoS Med 6(7): e1000097. doi:10.1371/journal.pmed1000097

**For more information, visit www.prisma-statement.org.**

**Fig 1. PRISMA flow diagram showing records identified by searching, screened, assessed for eligibility, and included in the systematic review.**

The schistosome-infected host exhibits a dynamic immune response to schistosome infection that is initially dominated by Th1 with some contributions from Th2, and then shifts over time towards Th2 and T regulatory (T reg) dominance during chronic infection [18,19]. The concomitant decrease in Th1 cytokines during this chronic phase underlies the hypothesis that the schistosome-infected host may develop impaired Th1 anti-viral and cytotoxic immune function. Most studies in this review examined immune effects of schistosome-virus coinfection during chronic schistosome infection. We examine each category of virus separately, first reviewing reported immune effects of schistosome-virus coinfection and then examining effects on viral prevalence/incidence, virulence, and vaccination for each virus.

## Hepatitis B and C viruses

**Effects of schistosome infection on host immune response to HCV.** In total, 15 studies described effects of schistosome infection on immune response to HCV and none focused on HBV (**Table 1**).

Six studies investigated circulating T cells in HCV-schistosome coinfected individuals compared to those with HCV alone [20–24]. In one of these, *S. mansoni*-infected patients who acutely acquired HCV developed a Th2 and T reg-dominant response and 17/17 failed to clear HCV, while those without *S. mansoni* infection developed a Th1-dominant response and 5/15 cleared HCV infection [20]. Both this study and another documented lower HCV-specific CD4[+] proliferative responses to HCV antigens in those with HCV-schistosome coinfection [20,21]. Two others reported no differences in CD4[+] or CD8[+] T cells, [22] or in CD4[+] or CD8[+] granzyme B[+] effector T cells, although granzyme B[+] effector T regs were increased in coinfection [23]. The fifth study reported fewer late differentiated HCV-specific memory T cells in coinfected individuals compared to those with HCV alone [24]. In the sixth study, *S. mansoni*-coinfected persons exhibited less frequent and smaller intrahepatic HCV-specific CD4[+] Th1 responses than those with HCV mono-infection, and biopsy-confirmed liver fibrosis was inversely correlated with the Th1 immune response [25].

Five studies evaluated host cytokine responses to HCV in schistosome-HCV coinfection versus HCV-monoinfection [23,26–29]. Two of these reported higher serum IL-4 and IL-10 and lower interferon-gamma (IFN-γ) levels in coinfected individuals [26,27], although two others did not [23,28]. Another reported higher IL-10 and lower IFN-γ in response to generic mitogen stimulation although not to HCV-specific stimulation [29]. Together, these studies generally support impaired Th1 anti-viral immunity in patients with schistosome-HCV coinfection.

Two studies investigated the role of HCV-schistosome coinfection in viral propagation in humans [30,31] and one investigated an analogous virus in mice [32]. In humans, HCV-infected blood samples cultured with *S. mansoni* [30] or *S. haematobium* [31] egg antigens had increased intracellular HCV replication and earlier detectable HCV RNA than HCV-infected blood samples cultured without schistosome egg antigens. Concordantly, the study in mice reported increased intrahepatic viral replication and reduced type 1 IFN production [32]. These findings are in agreement with data from some [23–25,28], but not all [27,29], studies that reported higher HCV RNA viral loads in patients with schistosome-HCV coinfection.

Two additional studies reported that *S. mansoni*-HCV coinfected patients had poorer response to interferon therapy than HCV-monoinfected patients, marked by more frequent relapses and poorer hepatic histologic improvement six months after therapy [33,34]. Studies assessing effects of schistosome infection with newer, direct-acting antivirals for HCV have not been performed. Together, this body of data consistently suggests impaired antiviral immune responses and poorer viral control in HCV-*S. mansoni* coinfection than HCV alone.

**Schistosomes and prevalence of HBV and HCV.** Studies on associations between schistosomes and HBV or HCV prevalence reported up to 2015 were previously summarized by

Table 1. Effects of Schistosome Coinfection on the Host Immune Response to Hepatitis C Virus.

| Study | Study Type | Sample Size | Sex (% Female) | Age in Years (Weighted Mean) | Country | Species | Immune and Viral Alterations in Schistosome Coinfected | *Overall Effects of Schistosomes on HCV Infection* |
|---|---|---|---|---|---|---|---|---|
| Kamal 2001 [20] | Human | 32 | 34% F | 28.0 | Egypt | S. mansoni | ↑ IL-4<br>↑ IL-10<br>↓ CD4⁺ | *Some studies report impaired anti-HCV responses of CD4+ helper T cells* |
| Kamal 2001 [21] | Human | 85 | 35% F | 38.0 | Egypt | S. mansoni | ↓ CD4⁺ | *Decreased Th1 immune response (IFN-γ) and increased Th2 immune response (IL-4, IL-10)* |
| Kamel 2014 [22] | Human | 82 | 43% F | 51.2 | Egypt | S. mansoni | No difference in CD4⁺ or CD8⁺ | *Increased HCV viral load and replication* |
| Loffredo-Verde 2015 [23] | Human | 44 | 41% F | 39.5 | Egypt | S. mansoni | No difference in CD4⁺ or CD8⁺<br>↑ granzyme B+ T regulatory cells<br>↑ HCV viral load | *Impaired response and increased relapse after interferon therapy* |
| Elrefaei 2003 [24] | Human | 24 | Not reported | 41.0 | Egypt | S. mansoni | ↓ Memory T-cells | |
| Kamal 2004 [25] | Human | 68 | 40% F | 34.3 | Egypt | S. mansoni | ↓ CD4⁺ Th1 intrahepatic T cells<br>↑ HCV viral load | |
| El-Kady 2005 [26] | Human | 70 | 30% F | 44.3 | Egypt | S. mansoni | ↑ IL-4<br>↑ IL-10<br>↓ IFN-γ | |
| Emam 2006 [27] | Human | 50 | 44% F | 44.7 | Egypt | S. mansoni | ↑ IL-4<br>↑ IL-10<br>↓ IFN-γ | |
| Ahmed 2016 [28] | Human | 107 | 47% F | Range, 30–81 years | Egypt | S. mansoni | ↑ HCV viral load<br>↑ IL-28B<br>No difference in IFN-γ | |
| Elrefaei 2004 [29] | Human | 38 | Not reported | 42.5 | Egypt | S. mansoni | No difference in IL-10 or IFN-γ with HCV peptide stim;<br>↑ IL-10 and ↓ IFN-γ with PHA stim* | |
| Bahgat 2010 [30] | *In vitro* | Not reported | Not reported | Not reported | Egypt | S. mansoni | ↑ HCV viral replication | |
| El-Awady 2006 [31] | *In vitro* | 26 | Not reported | Not reported | Egypt | S. mansoni | ↑ HCV viral replication | |
| Edwards 2005 [32] | Animal | Not reported | Not reported | Not reported | Not reported | S. mansoni | ↑ HCV viral replication | |
| El-Shazly 1994 [33] | Human | 60 | 17% | 41.0 | Egypt | S. mansoni | After interferon therapy:<br>↓ % ALT** normalization<br>↓ Response rate<br>↑ Relapse rate | |
| Kamal 2000 [34] | Human | 62 | 35% | 43.1 | Egypt | S. mansoni | After interferon therapy:<br>↓ Response rate<br>↑ Relapse rate | |

*Phytohemagglutinin (PHA) stimulation, a positive control

**Alanine aminotransferase

Abruzzi and colleagues [9] and therefore are not discussed in detail here. Briefly, schistosome infection is not associated with increased prevalence of HBV or HCV, aside from one study in Egypt where parenteral anti-schistosome treatment likely caused iatrogenic hepatitis infections. No additional studies of HBV or HCV prevalence or incidence that met inclusion criteria have been published since the 2015 review.

**Schistosomes and virulence of HBV and HCV.** Abruzzi and colleagues have meticulously summarized effects of schistosome-hepatitis virus coinfection on liver pathology [9], including several of the HCV immunology studies discussed above that also addressed virulence [20,21,25,27]. They concluded that studies in patients with schistosome-HCV coinfection indicate that schistosome-hepatitis coinfections have synergistic effects on liver disease. This was demonstrated by increased transaminases, cirrhosis, and/or portal hypertension in coinfected patients versus those with schistosomes or HCV alone. An additional HCV study published after the review also supports this finding [35].

Four studies in the Abruzzi review similarly reported that HBV-schistosome coinfected people had prolonged HbSAg antigenemia and greater liver fibrosis and splenomegaly compared to those with HBV alone. In a mouse model of HBV not included in the review, *S. mansoni* infection similarly promoted viral persistence, although liver pathology was decreased in coinfected compared to HBV-monoinfected mice during the T regulatory-dominant chronic phase of *S. mansoni* infection [36]. Interestingly, coinfected mice also cleared acute HBV infection more quickly during both the Th1-dominant acute phase of schistosome infection and the chronic T regulatory dominant phase [36]. These studies demonstrate the important role that timing of infections may have on viral virulence and the need for human data to clarify these findings.

Two small studies examined effects of praziquantel treatment on HBV disease course. In ten *S. mansoni*-HBV co-infected individuals, HBsAg remained positive in the four individuals with persistent *S. mansoni* six months post-treatment, versus in only one of six who were parasitologically cured [37]. The other study reported that praziquantel treatment of HBV-*S. mansoni* coinfected individuals led to clearance of HbSAg in 20% and normalization of liver transaminases in ~15% after 6 months, while those with HBV alone who were followed for 6 months had no clearance of HbSAg and ~10% developed abnormal transaminases [38].

**Schistosomes and HBV vaccination.** Eight studies investigated hepatitis B vaccine immunogenicity with and without schistosome coinfection (**Table 2**). Mice with chronic *S. japonicum* had significantly attenuated anti-hepatitis B surface (HBs) antibody response to hepatitis B vaccine, and anti-HBs antibody titers increased after praziquantel treatment [39]. Two studies reported lower post-vaccine anti-HBs titers in *S. mansoni*-infected children [40] and adults [41]. Titers in *S. mansoni*-infected adults were lower two weeks after the second hepatitis B vaccine dose, with a trend towards remaining lower after three doses [41]. Another study involving 504 children found no impact of *S. mansoni* infection on anti-HBs antibody titers when children were given praziquantel treatment and vaccinated concurrently [42]. These data suggest that hepatitis B vaccine response may be compromised by schistosome infection and improved after treatment. Notably, four studies reported that maternal schistosome infection does not impair neonatal response to hepatitis B vaccine [43–46].

Collectively, these data demonstrate that schistosome infection is associated with impaired anti-viral immune responses and subsequent increased viral replication and liver damage. This has been particularly well studied in adults with *S. mansoni* and HCV; very little is known in children who have acquired HBV perinatally. Schistosome infection does appear to depress the humoral HBV vaccine response in children. Importantly, praziquantel may aid viral control, decrease liver pathology, and improve HBV vaccine response and could be an effective means to decrease global burden of these viral infections in schistosomiasis endemic regions.

## HIV

**Effects of schistosome infection on host immune response to HIV.** Three studies investigated the mucosal anti-HIV immune defense in HIV-uninfected women (**Table 3**). One study documented associations of *S. haematobium* infection with higher frequencies of CD14$^+$

**Table 2. Effects of Schistosome Infection on Response to Hepatitis B Vaccination.**

| Study | Subject Type | Sample Size | Sex (% Female) | Age in Years (Weighted Mean) | Country | Species† | Vaccine Response in Schistosome Infected | *Overall Effects of Schistosomes on HBV Vaccine* |
|---|---|---|---|---|---|---|---|---|
| Chen 2012 [39] | Mouse | 140 | 0% F | N/A | N/A | *Sj* | ↓ **post-vaccine anti-HBs titers**\*\* **Titers comparable to uninfected after praziquantel** | *Lower post-vaccine anti-hepatitis B surface antigen antibody titers.* |
| Ghaffar 1990 [40] | Human | 80 | 0% F | Range, 8–12 years | Egypt | *Sm* | **Mean anti-HBs titers 68 mIU/mL in *Sm* versus 335 in controls**\*\* | *Vaccine responsiveness may improve after praziquantel treatment.* |
| Riner 2016 [41] | Human | 146 | 53% F | 21 | Kenya | *Sm* | **Mean anti-HBs titers 158 mIU/mL in *Sm* versus 561 in controls**\* | |
| Bassily 1992 [42] | Human | 508 | 40% F | Range, 6–12 years | Egypt | *Sm* | **No difference in titers in *Sm* infected and uninfected Fewer children with hepatosplenomegaly had adequate titers**\*\* | |
| Bassily 1997 [43] | Human | 385 | 49% F | Infants followed to age 9 months | Egypt | *Sm* | **No difference in seroconversion or mean anti-HBs titers in infants of mothers with/without *Sm*** | *Maternal schistosome infection does not seem to impact infant response to hepatitis B vaccine.* |
| Nash 2017 [44] | Human | 1379 | 50% F | Infants followed to age 1 year | Uganda | *Sm* | **No effect of maternal praziquantel treatment on infant anti-HBs titers.** | |
| Malhotra 2015 [45] | Human | 450 | Not reported | Infants followed to age 36 months | Kenya | *Sh* | **No effect of maternal *Sh* infection on infant anti-HBs titers.** | |
| Malhotra 2018 [46] | Human | 450 | Not reported | Infants followed to age 30 months | Kenya | *Sh* | **Prenatal *Sh* exposure did not affect infant anti-HBs titers or trajectories of titers.** | |

†*Sm = S. mansoni; Sh = S. haematobium; Sj = S. japonicum*

\*p<0.10

\*\*p<0.05

and CD4$^+$ cells expressing the HIV coreceptor CCR5 in the genital tract and systemic circulation, which decreased after praziquantel treatment [47]. An in-vitro study found that HIV viral entry into cervical and systemic CD4$^+$ cells was higher in *S. mansoni*-infected women and decreased after praziquantel treatment [48]. Treatment of *S. mansoni* infection also partially normalized dysregulated anti-viral interferon signaling [48]. This corresponds with observed lower levels of anti-viral cytokine interleukin-15 in cervicovaginal lavages of *S. haematobium*-infected women [49]. By contrast, immune cell populations were not altered in the foreskins of men with *S. mansoni* infection [50]. Taken together, these consistently report that schistosome-infected women have genital mucosal alterations in immune cell populations, anti-viral signaling, and HIV viral entry. Together with physical breaches of the mucosal barrier by schistosome eggs, these may contribute to increased susceptibility to mucosal HIV acquisition in women.

Five additional human studies examined host systemic immunity in HIV-schistosome coinfected patients, and all reported increased Th2 or Treg responses in those with coinfection versus those with HIV alone [50–54]. HIV-schistosome coinfected men had higher densities of HIV co-receptors CXCR4 and CCR5 on the surface of CD4$^+$ T cells, which were upregulated by Th2/Treg cytokines IL-4 and IL-10 [51]. Higher systemic frequencies of HIV-susceptible Th1, Th17, and Th22 CD4$^+$ T cell subsets were also observed in HIV-uninfected men with *S. mansoni* [50]. Stimulation of PBMCs with HIV-1 Gag peptides yielded decreased Gag-specific CD8$^+$ cytolytic T-cell responses and increased Gag-specific IL-10-positive CD8$^+$ T-cell responses in HIV-*S. mansoni* coinfected compared to HIV-monoinfected individuals, suggesting dysregulated cellular immunity to HIV [52]. These five studies thus demonstrate

**Table 3. Effects of Schistosome Infection on Host Immune Response to HIV in the Genital Mucosa and Systemic Circulation.**

| Study | Sample Size | Sex (% Female) | Age in Years (Weighted Mean) | Country | Species† | Immune Alterations in Schistosome Coinfected | Overall Effects of Schistosome on HIV Infection |
|---|---|---|---|---|---|---|---|
| **Genital Mucosal Immune Effects** | | | | | | | |
| Kleppa 2014 [47]* | 44 | 100% F | 18 | South Africa | Sh | ↑ cervical HIV target cells & coreceptors (CD14⁺, CD4⁺CCR5⁺) | Genital mucosal immune alterations are potentially consistent with increased HIV susceptibility in women. |
| Yegorov 2019 [48]* | 36 | 100% F | 24.5 | Uganda | Sm | ↑ HIV viral entry into cervical CD4⁺ cells ↑ IFN-α after schistosome treatment | |
| Dupnik 2019 [49] | 97 | 100% F | 28.9 | Tanzania | Sh, Sm | ↓ Anti-viral cytokine (IL-15) in Sh but not Sm | |
| Prodger 2015 [50]* | 34 | 0% F | 24.6 | Uganda | Sm | No differences in frequencies or activation of HIV target cells in foreskin | |
| **Systemic Circulation Immune Effects** | | | | | | | |
| Kleppa 2014 [47]* | 44 | 100% F | 18 | South Africa | Sh | ↑ blood HIV target cells & coreceptors (CD14⁺, CD4⁺CCR5⁺) | Peripheral blood immune alterations are potentially consistent with increased HIV susceptibility in both men and women. |
| Yegorov 2019 [48]* | 36 | 100% F | 24.5 | Uganda | Sm | ↑ HIV viral entry into blood CD4⁺ cells | |
| Prodger 2015 [50]* | 34 | 0% F | 24.6 | Uganda | Sm | ↑ frequencies of Th1, Th17, and Th22 CD4⁺ T cell subsets in blood | |
| Secor 2003 [51] | 42 | 0% F | Age > 18 years | Kenya | Sm | ↑ density of CXCR4 on CD4⁺ T cells and CD14⁺ monocytes | |
| McElroy 2005 [52] | 35 | Not reported | 36.6 | Uganda | Sm | ↓ HIV Gag-specific CD8⁺ cytolytic T-cell responses ↑ Gag-specific IL-10-positive CD8⁺ T-cell responses | |
| Erikstrup 2008 [53] | 378 | 80% F | 33 | Zimbabwe | Sh, Sm | ↑ IL-8 in Sm but not Sh ↓ TNF-α receptor and IL-8 after schistosome treatment | |
| Obuku 2016 [54] | 50 | At least 28% F | 29 | Uganda | Sm | ↑ TNF-α after non-viral stimulation ↑ CD8⁺IFN-γ⁺ and CD4⁺IFN-γ⁺TNF-α⁺ T cells after Gag stimulation | |
| Mouser 2019 [63] | N/A–in vitro study | N/A | N/A | N/A | Sm | ↓ trans-HIV infection after Sm antigen stimulation in dendritic cells that matured in Th2 conditions | |
| **Animal Studies** | | | | | | | |
| Dzhivhuho 2018 [55] | Not reported | Mice, 100% F | N/A | N/A | Sm | ↓ IFN-γ and IL-2 in response to challenge with candidate HIV vaccines | Animal models support human studies, demonstrating potentially impaired immunity to HIV. |
| Actor 1993 [56] | Not reported | Mice, not reported | N/A | N/A | Sm | ↓ IFN-γ and IL-2 in response to challenge with vaccinia virus expressing gp160 ↓ CD8⁺ anti-viral cytotoxicity ↓ viral clearance | |
| Marshall 2001 [57] | Not reported | Mice, 100% F | N/A | N/A | Sm | ↓ CD8⁺ anti-viral cytotoxicity related to a splenic immune cell population | |
| Actor 1994 [58] | Not reported | Mice, not reported | N/A | N/A | Sm | ↓ anti-viral cytotoxicity and vaccinia viral clearance began 6 weeks after Sm infection, associated with egg granuloma formation | |
| Lacroix 1998 [59] | Not reported | Mice, 100% F | N/A | N/A | Sm | ↑ IL-4 in mice with Sm-murine HIV coinfection | |
| Chenine 2005 [60] | 8 | Macaques, not reported | N/A | N/A | Sm | ↑ IL-4 and IL-10 in Sm ↑ sHIV viral loads in Sm | |
| Ayash-Rashkovsky 2007 [61] | 15 | Macaques, not reported | N/A | N/A | Sm | ↓ CD4⁺CD29⁺ memory T cells in Sm ↑ IL-4 in Sm ↑ reactivation of previously undetectable sHIV in Sm | |
| Buch 2001 [62] | 8 | Macaques, not reported | N/A | N/A | Sm | ↑ sHIV replication in tissue macrophages in Sm accompanied by ↑ IL-4 and ↓ IFN-γ | |

* indicates study examined both genital mucosa and systemic circulation

†Sm = S. mansoni; Sh = S. haematobium; Sj = S. japonicum

schistosome-associated alterations in host systemic immunity that could confer increased susceptibility to HIV.

Notably, some upregulations in pro-inflammatory markers IL-8, TNF receptor sTNF-rII, and TNF-α have been documented in schistosome-HIV coinfection, possibly consistent with heightened Th1 response following recent HIV acquisition [53,54]. However, overall, these human studies report that schistosome coinfection downregulates Th1 and cytotoxic immune responses and upregulates Th2 and Treg responses in HIV-infected individuals, and that treating the schistosome infection modulates these differences.

Eight animal studies reported decreased Th1 and increased Th2/T reg responses to HIV during schistosome infection [55–62]. In the first, splenocytes from *S. mansoni*-infected mice stimulated with HIV vaccine candidates produced less IFN-γ and IL-2 than *S. mansoni*-uninfected, vaccinated mice, and partial restoration was observed after anti-schistosome treatment [55]. In two other studies, mice with vaccinia-*S. mansoni* coinfection produced minimal IFN-γ and IL-2 in response to viral challenge with recombinant vaccinia virus expressing the HIV glycoprotein gp160, and exhibited delayed viral clearance, compared to mice with vaccinia alone [56,57]. Longitudinal mouse studies documented delayed viral clearance only after the onset of worm egg-laying, correlating with decreasing IFN-γ and IL-2, impaired viral-specific cytotoxic CD8+ T-cell responses [58], and increased IL-4 [59]. The Th2 predominance after egg-laying further links the impaired anti-viral response during schistosome infection to dwindling Th1 immune responses at the time of worm maturation and oviposition [63]. Elevated Th2/T reg cytokines and decreased IFN-γ have also been reported in macaques with sHIV-schistosome coinfection and have been linked to increased sHIV replication in macrophages [60–62]. Together, these animal studies support the human findings of Th2 predominance and potentially impaired anti-HIV immunity during chronic schistosome infection.

Finally, an *in vitro* study documented increased resistance to trans-infection in HIV-susceptible human dendritic cells that matured while exposed to schistosome egg antigens in a Th2-predominant environment compared to cells that matured in a Th1 environment [64]. This suggests that during schistosome-HIV coinfection, HIV may spread less readily from cell-to-cell via *trans*-infection than during HIV mono-infection, indicating a potential means by which schistosome coinfection, perhaps unexpectedly, could slow viral replication and disease progression.

**Schistosomes and prevalence and incidence of HIV.** Cross-sectional studies investigating HIV prevalence in schistosome-infected and -uninfected individuals have been inconclusive (**Table 4**). Four studies reported higher prevalence of HIV in schistosome-infected versus uninfected women [65–68], of which three focused on *S. haematobium* infection. Five others, conducted in *S. mansoni*-hyperendemic regions or in men only, found no difference by schistosome status [69–73], although at least one was under-powered.

Five studies investigated effects of schistosome infection on HIV incidence. In a nested case-control study in Uganda, investigators found no increased incidence of HIV in *S. mansoni*-infected people, although the odds of incident HIV was eight-fold higher in those who had not received anti-schistosome treatment in the past two years [74]. Increased incidence of HIV was similarly not observed in serodiscordant couples and sex workers in Uganda and Kenya who had either *S. haematobium* or *S. mansoni* infection [75]. In a cohort in Tanzania, persons who had *S. haematobium* eggs in urine at any point during the study did not have increased HIV incidence [76]. In contrast, a nested case-control study in Tanzania reported increased odds of HIV acquisition (OR = 2.8 [95% confidence interval, 1.2–6.6]) in schistosome-infected women but not men [77]. One additional study reported no increased mother-to-child HIV transmission from schistosome-infected mothers, although infection with any helminth (including schistosomes) strongly increased odds of HIV transmission suggesting

**Table 4. Prevalence and Incidence of HIV Among Schistosome-Infected and Uninfected Individuals.**

| Study | Subject Type | Sample Size | Sex (% Female) | Age in Years (Weighted Mean) | Country | Species† | Reported Prevalence/Incidence | Overall Effects of Schistosome Infection |
|---|---|---|---|---|---|---|---|---|
| **HIV Prevalence–Cross-Sectional Studies** | | | | | | | | |
| Kjetland 2006 [65] | Human | 527 | 100% F | 34 | Zim-babwe | *Sh* | **HIV prevalence: 41% among *Sh* 26% among uninfected*** | *Increased prevalence of HIV has been reported among women with Sh or mixed Sh/Sm infections.* |
| Downs 2011 [66] | Human | 457 | 100% F | 30 | Tanzania | *Sh Sm* | **HIV prevalence: 17% among *Sh* 6% among uninfected*** | |
| Downs 2012 [67] | Human | 345 | 100% F | 30 | Tanzania | *Sh and Sm* | **HIV prevalence: 9% among *Sh* or *Sm* 3% among uninfected*** | |
| Mvumbi 2018 [68] | Human | 446 | 100% F | Not reported | Demo-cratic Republic of Congo | *Sh* | **HIV prevalence: 28% among *Sh* 11% among uninfected*** | |
| Downs 2017 [69] | Human | 674 | 0% F | 34 | Tanzania | *Sh and Sm* | **HIV prevalence: 6.5% among *Sh* 6.4% among uninfected 5% among *Sm* 6% among uninfected** | *No difference has been reported in HIV prevalence in men for Sh or Sm.* |
| Wood-burn 2009 [70] | Human | 2507 | 100% F | Median: 23 | Uganda | *Sm* | **Odds of HIV in *Sm* infected: 1.06 (95% CI 0.78–1.51)** | *No difference has been reported in HIV prevalence in many Sm studies.* |
| Sanya 2015 [71] | Human | 1412 | 45% F | 30.3 | Uganda | *Sm* | **HIV prevalence: 21% among *Sm* 18% among uninfected** | |
| Mazigo 2014 [72] | Human | 1785 | 52.9% F | 35.6 | Tanzania | *Sm* | **HIV prevalence: 6% among *Sm* 8% among uninfected** | |
| De Lima e Costa 1988 [73] | Human | 180 | Not reported | 20.5 | Brazil | *Sm* | **HIV prevalence: 3.6% among *Sm* 0 among uninfected** | |
| **HIV Incidence–Longitudinal Studies** | | | | | | | | |
| Ssetaala 2015 [74] | Human | 200 | Not reported | Range: (13–49) | Uganda | *Sm* | ***Sm* prevalence was 49% among HIV-seroconverters and 52% among controls** | *Several studies have shown no increased risk of HIV acquisition with Sm.* |
| Bochner 2020 [75] | Human | 2250 | 75% F | ~31# | Kenya & Uganda | *Sh and Sm* | ***Sh/Sm* prevalence was 27% among HIV-seroconverters and 25% among controls Trend toward increased risk for *Sh* in women (36% versus 25%).** | *Some studies suggest increased risk of HIV acquisition in women, more strongly with Sh.* |
| Kroidl 2016 [76] | Human | 1055 | 51% F | 15.4 | Tanzania | *Sh* | **HIV incidence per 100 person-years: 1.3 among *Sh* 1.2 among uninfected** | |
| Downs 2017 [77] | Human | 338 | 61% F | Not reported | Tanzania | *Sh and Sm* | **Women: *Sh/Sm* prevalence was 43% among HIV-seroconverters and 30% among controls* Men: 29% among HIV-seroconverters and 38% among controls** | |
| Galla-gher 2005 [78] | Human | 936 | 100% F | Not reported | Kenya | *Sh* | ***Sh* prevalence was 17% among mothers who transmitted HIV to children and 9% among mothers who did not [85% versus 40% for any helminth]*** | |
| Chenine 2008 [79] | Animal | 17 | 100% F | N/A | N/A | *Sm* | **17-fold lower sHIV viral dose needed to achieve rectal transmucosal infection** | *Animal models support increased mucosal sHIV sus-ceptibility.* |
| Siddappa 2011 [80] | Animal | 16 | 100% F | N/A | N/A | *Sm* | **No difference in sHIV dose needed to achieve intravenous infection** | |

† *Sh = S. haematobium; Sm = S. mansoni; Sj = S. japonicum*

* Difference was statistically significant.

# Approximate calculation; data was incomplete to determine precise weighted mean.

the study may have been underpowered to examine individual helminths [78]. Additional challenges with this body of evidence include lack of control for confounders including sexually-transmitted infections in high-risk cohorts [75], unclear timing of anti-schistosome

treatment in relation to HIV acquisition [76], and use of assays unable to differentiate between schistosome species [77].

Studies in macaques more clearly support increased HIV susceptibility during schistosome infection. Investigators reported that *S. mansoni*-infected macaques, when rectally inoculated with increasing doses of simian HIV (sHIV), became infected at viral doses 17-fold lower than schistosome-uninfected macaques [79]. The same investigators found no difference in sHIV dose needed to achieve infection when schistosome-infected and uninfected macaques were infected intravenously, suggesting that the sHIV susceptibility is mediated by mucosal effects of *S. mansoni* [80]. Because macaques received rectal sHIV inoculation, *S. mansoni* in these studies may mimic the effects of *S. haematobium* infection on vaginal HIV exposure in humans. Although parasite eggs can be found in the genital tract in both *S. haematobium* and *S. mansoni* infections, they are more highly concentrated and associated with greater genital tract pathology in *S. haematobium* [81–84].

Overall, this body of evidence is inconclusive but suggests that schistosome infection may increase the risk for HIV acquisition in women, with a stronger signal toward increased risk with *S. haematobium*. The evidence for *S. mansoni* is mixed and an association with HIV, if present, may be lower in magnitude. This is corroborated by a third, larger nested case-control study, not eligible for inclusion in the review due to its use of antibody testing, which reported a moderately increased hazard of HIV acquisition in women with anti-*S. haematobium* antibodies and a smaller increased hazard for women with anti-*S. mansoni* antibodies that did not reach significance (hazard ratios = 1.44 [1.05–1.96] and 1.30 [0.91–1.87], respectively) [85].

Interestingly, that case-control study also indicated that the presence of anti-schistosome antibodies was associated with increased transmission of HIV from HIV-infected men and women to their sexual partners, as well as with increased hazard of death in HIV-infected women [85]. Men with *S. haematobium* infection have also been reported to have an increased concentration of leukocytes and eosinophils in semen, cell types that are permissive to HIV replication and may facilitate onward HIV transmission from men with HIV-*S. haematobium* coinfection to their sexual partners [86]. Further supporting this possibility, a small study of 6 HIV-*S. haematobium* coinfected men who were not on ART and were treated with praziquantel reported a decrease in HIV-1 RNA viral load in semen from baseline to 10 weeks post-treatment, though this did not reach significance [87].

**Schistosomes and HIV virulence.** One study reported that *S. mansoni*-HIV coinfection was associated with elevated liver transaminases after controlling for other known causes of transaminitis (HBV, HCV, alcohol, medications) in patients on ART [88]. This suggests that *S. mansoni* infection could potentiate direct hepatotoxicity of HIV on hepatocytes [89].

Thirteen studies examined effects of HIV-schistosome coinfection on CD4$^+$ T cell counts (**Table 5**). Seven found no relationship [54,72,90–94], one reported higher CD4$^+$ counts [95], and one reported lower increase in CD4$^+$ counts following ART initiation in coinfected patients [96]. Four other studies found no difference in CD4$^+$ counts before and 12 weeks after praziquantel treatment [95,97,98] or in those given praziquantel quarterly versus annually [99]. Another study reported that individuals successfully treated for schistosome infection had lower CD4$^+$ counts after six months than those still infected [100]. Taken together, these studies indicate that schistosome infection does not consistently affect CD4$^+$ count, and that treatment of schistosome infection seems not to increase, and could possibly decrease, CD4$^+$ counts.

Seven studies examined schistosome infection and HIV-1 viral load. In adults who had recently acquired HIV, median viral load was higher in those with HIV-schistosome coinfection than with HIV alone in one study [77] but lower in a combined analysis of four cohort studies in higher-risk populations [101]. Cervical HIV-1 viral loads were also unexpectedly

**Table 5. Effects of Schistosome-HIV Coinfection on CD4+ T Cell Counts, HIV-1 RNA Viral Load, and HIV Disease Progression.**

| Schistosome HIV Co-Infection Effects on CD4 Count | | | | | |
|---|---|---|---|---|---|
| **No relationship** | **Higher CD4 counts** | **Lower increase in CD4+ counts following ART initiation** | **No difference before and after praziquantel** | **No difference whether praziquantel is quarterly or annual** | **Lower CD4+ counts 6 months after schistosomiasis treatment** |
| Obuku 2016 [54] | Elliott 2003 [95] | Efraim 2013 [96] | Elliott 2003 [95] | Abaasa 2018 [99] | Brown 2005 [100] |
| Mazigo 2014 [70] | | | | | |
| Brown 2004 [90] | | | | | |
| Kleppa 2015 [91] | | | Mulu 2013 [97] | | |
| Colombe 2018 [92] | | | | | |
| Colombe 2018 [93] | | | Mazigo 2016 [98] | | |
| Idindili 2011 [94] | | | | | |

| Schistosome Infection and HIV-1 Viral Load | | | | | | |
|---|---|---|---|---|---|---|
| **Higher viral load** | **Lower viral load** | **No difference in viral load** | **Higher viral load, declining over time** | **No difference before and after praziquantel** | **Transient increase in viral load following praziquantel** | **Higher viral load >6 months after praziquantel** |
| Downs 2017 [77] *HIV recently acquired* | Bochner 2019 [101] *HIV recently acquired* | Masikini 2019 [102] *Patients failing first-line ART* | Chenine 2008 [79] *Animal Study* | Brown 2004 [90] | Elliott 2003 [95] | Lawn 2000 [104] *Pre-ART study* |
| Brown 2004 [90] *Chronic HIV* | Colombe 2018 [92] *Chronic HIV* | | | Kallestrup 2005 [103] *Pre-ART study; Also, viral loads increased pre-treatment those whose praziquantel was delayed* | Brown 2005 [100] | Abaasa 2018 [99] *Patients not yet eligible for ART* |
| Mazigo 2016 [98] *Chronic HIV* | Elliott 2003 [95] *Chronic HIV* | | | Mazigo 2016 [98] | | |

| Slower HIV Disease Progression in those with Chronic HIV-Schistosome Coinfection | | |
|---|---|---|
| Colombe 2018 [93] *Patients not yet on ART* *Outcome: CD4+ count < 350 cells/μL or death* *Hazard ratio = 0.31 [0.12–0.84]* | Abaasa 2018 [99] *Patients not yet eligible for ART* *Outcome: progression to AIDS* *Log-rank chi-square (low-intensity versus high-intensity praziquantel) = 2.08, p = 0.15* | Stete 2018 [105] *Patients starting ART* *Outcome: death or loss-to-follow-up* *Hazard ratio = 0.58 [0.32–1.05]* |

lower in *S. haematobium*-coinfected patients [101]. In four studies among patients with advanced HIV infection, schistosome-HIV coinfected individuals had lower viral loads in one [92]. higher viral loads in one [98], and no significant differences in two [90,95]. Another study reported no difference in viral loads by schistosome status among patients failing first-line ART [102]. These discrepant results suggest a complex relationship between schistosome infection and HIV-1 viral load that may be influenced by duration of HIV infection. Macaques with *S. mansoni* infection also had higher HIV-1 viral loads shortly after mucosal HIV acquisition, which declined over time [79], though macaques naturally clear *S. mansoni* infection after several months and this precludes definite conclusions [60].

Examination of the effects of praziquantel treatment on viral load provides additional insight. In six studies of viral load in HIV-schistosome coinfected individuals within six months of praziquantel treatment, two reported no difference before and after treatment

[90,98]. One small randomized trial reported higher viral loads in HIV-schistosome coinfected persons randomized to receive delayed treatment after three months than in persons who received praziquantel at baseline [103]. Two other studies reported increases in viral loads 4–5 weeks after treatment, which were not sustained when re-measured 4–5 months after treatment [95,100]. The transient increase in viral load was especially prominent in patients with high-intensity schistosome infection at baseline [100]. Together, these consistently report increased HIV-1 viral load shortly after praziquantel treatment, which decreased within three months. Notably, these studies neither categorized people by treatment success or failure, nor controlled for other possible causes of viral load elevations.

In contrast, two studies examined effects of praziquantel treatment over periods longer than six months and assessed for schistosome eradication. One pre-ART study reported increased viral load after effective treatment, particularly six or more months post-treatment [104]. In the other study, schistosome-infected persons not yet eligible for ART who received quarterly praziquantel treatment also tended towards higher viral loads, as compared to controls who received praziquantel once and among whom 73% remained infected one year post-treatment (mean viral load ratio 1.88 [0.78, 4.53] p = 0.16) [99]. Therefore, these two studies suggest, rather unexpectedly, that long-term successful treatment of schistosome coinfection could increase HIV-1 viral load.

Consistent with these findings, three longitudinal studies have suggested slower HIV disease progression in those with chronic HIV-schistosome coinfection, versus those with HIV infection alone—results that contradicted investigators' original hypotheses. In the first study, people with schistosome coinfection not yet on ART had a lower hazard ratio of developing CD4$^+$ counts <350 cells/μL or death (HR = 0.31 [0.12–0.84]) [93]. The study of quarterly praziquantel treatment described above also reported a trend toward increased risk of progression to AIDS in the intensively-treated, ART-naïve group (log-rank chi-square 2.08, p = 0.15) [99]. A third study reported a hazard ratio of 0.58 [0.32–1.05] for death or loss to follow-up in those with HIV-schistosome coinfection starting ART, almost reaching significance [105].

Differences in results reported in these studies highlight the complexity of the relationship between HIV-1 viral load and schistosome infection, which may vary by schistosome species, other coinfections, and parasite and/or viral dynamics. One possible hypothesis to harmonize these studies is that schistosome and HIV infections could interact differently over time, with HIV-1 viral load set-points elevated shortly after HIV acquisition in some groups of people with schistosome coinfection, and later becoming lower in coinfected people as HIV becomes chronic. Taken together, these data raise the possibility that, in people with chronic HIV-schistosome co-infection, viral loads may be lower and disease progression slower than in chronic HIV alone.

**Schistosomes and HIV vaccination.** Three murine studies reported lower magnitude of HIV-specific CD8$^+$ T-cell responses to HIV vaccines in schistosome-infected than uninfected mice (**Table 6**) [55,106,107]. In one of these, mice given praziquantel two weeks before vaccination still had lower titers of gp140-specific IgG antibodies and only partial restoration of CD4$^+$ and CD8$^+$ T-cell responses 12 days after vaccination [55]. Further, the presence of schistosome eggs in tissue, even in the absence of active worm infection, attenuated anti-gp140 antibody responses [55]. Another study reported that *S. mansoni* infection attenuated HIV-specific CD8$^+$ IFN-γ secretion [106]. Praziquantel treatment four weeks before vaccination caused partial restoration of this CD8$^+$ response, and treatment eight weeks before vaccination completely restored this response [107]. These studies collectively demonstrate that schistosome infection could decrease CD8$^+$ T-cell responses to an HIV vaccine, and that restoration of CD8$^+$ responses occurs at least eight weeks after treatment, the approximate time over which the burden of eggs trapped in tissue decreases [108].

Table 6. **Effects of *Schistosoma mansoni* Infection on Response to HIV Vaccine in Mice.**

| Study | Study Design | Time Elapsed Between Vaccine and Assessment | Vaccine Response in Schistosome Infected | *Overall Effects of S. mansoni on HIV Vaccine* |
|---|---|---|---|---|
| Dzhivhuho 2018 [55] | Mice with and without *Sm* were given candidate HIV vaccines. Some mice were treated with PZQ* pre-vaccine | 12 days post-vaccine (vaccinated 2 weeks after PZQ) | ↓ HIV-specific splenocyte IFN-γ and IL-2 secretion. Partial restoration of IFN-γ and IL-2 after PZQ. ↓ gp-140 IgG antibody production in *Sm* | *Impaired HIV-specific cellular immune response to HIV vaccine* |
| Da'dara 2006 [106] | Mice with and without *Sm* were given candidate HIV vaccines | 2 weeks | ↓ HIV-specific IFN-γ secretion by CD8+ T cells in *Sm* | *Cellular immune response was restored completely when PZQ treatment was provided at least 8 weeks prior to vaccination* |
| Da'dara 2010 [107] | Mice with *Sm*, mice with prior *Sm* who were treated, and uninfected mice were given candidate HIV vaccines | 2 weeks post-vaccine (vaccinated 4 or 8 weeks after PZQ) | ↑ HIV-specific IFN-γ secretion by CD8+ T cells after PZQ (completely restored when vaccinated 8 weeks after PZQ) | |

*PZQ = praziquantel

## Herpesviruses in mice and humans

Herpesviruses, which cause lifelong infection marked by periods of latency and reactivation, may offer important insights into interactions between schistosomes and viruses. A study in mice with chronic latent murine γ-herpesvirus that were infected with *S. mansoni* developed systemic herpesvirus reactivation with concomitant increases in IL-4 and decreased IFN-γ compared to herpesvirus mono-infected mice (**Table 7**) [109].

In China, men with *S. japonicum* infection had higher KSHV seroprevalence compared to men without *S. japonicum*, although no difference was observed in women [110]. Similarly, no difference in KSHV seroprevalence was observed in pregnant women in Uganda based on *S. mansoni* status [111]. In contrast, in Ugandan fishing communities, seroprevalence of KSHV was 89% in those with *S. mansoni* infection versus 77% in those without, and odds of KSHV remained significant after adjusting for sex, age, and other parasitic infections [112]. In addition, elevated anti-K8.1 antibodies in *S. mansoni* infection suggested increased KSHV reactivation [112]. These studies, supported by mouse models discussed previously [109], suggest that

Table 7. **Effects of Schistosome Infection on the Host Immune Response to Herpesviruses, and on Prevalence of Kaposi's Sarcoma-Associated Herpesvirus.**

| Study | Subject Type and Virus | Sample Size | Sex (% Female) and Age | Country | Species† | Differences in Schistosome Infected Compared to Uninfected | Overall Effects of Schistosome Infection |
|---|---|---|---|---|---|---|---|
| Reese 2014 [109] | Mice; murine γ-herpes-virus 68 | Not reported | Not reported | N/A | *Sm* or *Hp* | **↑ systemic herpes-virus reactivation. Reactivation was dependent on IL-4 and STAT-6** | *Increased Th2 cyto-kines were associated with systemic viral reactivation.* |
| Fu 2012 [110] | Human, KSHV | 105 | 47% F, Age range not reported | China | *Sj* | **KSHV prevalence: 8.4% among *Sj* 6.6% among uninfected (men: 8.4% versus 2.2%)*** | *Increased KSHV sero-prevalence in Sm, with evidence stronger for men.* |
| Wakeham 2011 [111] | Human, KSHV | 1915 | 100% F, median age 23 | Uganda | *Sm* | **KSHV prevalence: 60% among *Sm* 61% among uninfected** | |
| Nalwoga 2019 [112] | Human, KSHV | 1137 | 44% F, median age 25 | Uganda | *Sm* | **KSHV prevalence: 89% among *Sm* 77% among uninfected* Odds increased after controlling for sex*** | |

†*Sm = S. mansoni; Sh = S. haematobium; Sj = S. japonicum; Hp = Heligmosomoides polygyrus*
*Statistically significant

KSHV is an additional virus for which schistosome infection may impair host anti-viral defense and subsequent control, with stronger evidence for this effect in men than in women.

## Respiratory viruses

**Effects of schistosome infection on host immune response to respiratory viruses.** Five studies have investigated immune effects of *S. mansoni* infection on respiratory viruses in mice (**Table 8**). In the first study, mice with acute respiratory herpesvirus-*S. mansoni* coinfection had increased IL-4 and expanded CD8$^+$ T-cell viral-specific effector responses in the lung, leading to enhanced control of the respiratory virus [113]. This suggests that schistosome-induced Th2 predominance, demonstrated here by increased IL-4, may be beneficial for viral control in the lung.

A similar beneficial anti-viral effect in the lung has been observed in mice coinfected with *S. mansoni* plus influenza A. Mice with *S. mansoni* infection that were challenged with influenza A virus had decreased mortality, decreased weight loss, decreased alveolar inflammation, lower expression of pro-inflammatory cytokines IL-6 and TNF-α, and greater lung epithelial regeneration than mice without *S. mansoni* [114–116]. Comparable beneficial effects have been observed in mice with *S. mansoni* and pneumonia virus [117] and in studies of other helminths [118]. Proposed mechanisms for protection include goblet cell hyperplasia, excessive mucus secretion, and alterations in interferon levels. These intriguing data may be attributable to the unique environment of the lung compared to the rest of the body. Together with the studies of schistosome infection during chronic HIV, these mouse respiratory virus data exemplify another scenario in which schistosome-viral coinfection may be surprisingly beneficial compared to viral infection alone.

**Table 8. Effects of Schistosome Infection on the Host Immune Response to Respiratory Viruses in Mice.**

| Study | Virus | Sex (% Female) | Species† | Differences in Schistosome Infected Compared to Uninfected | Overall Effects of Schistosome Infection |
|---|---|---|---|---|---|
| Rolot 2018 [113] | Murine respiratory γ-herpes-virus 4 | 100% F | *Sm* | In lung:<br>↑ IL-4<br>↑ CD8$^+$ T-cell viral-specific effector responses<br>↓ virus titers | *Increased Th2 responses and decreased Th1 responses were associated with better viral control in the lung.* |
| Danz 2016 [114] | Influenza A | 100% F | *Sm* | ↓ mortality and morbidity<br>↓ influenza-specific IFN-γ secretion by mediastinal CD8$^+$ T cells<br>↓ IFN-γ transcripts in lungs<br>↓ alveolar inflammation | |
| Tundup 2017 [115] | Influenza A | Not reported | *Sm* | ↓ weight loss and lung injury<br>↑ dendritic cells and alternatively activated macrophages in lungs<br>↓ expression of lung IL-6, TNF-α | |
| Kadl 2018 [116] | Influenza A | Not reported | *Sm* | ↓ mortality<br>↑ influenza-specific IgM & IgG<br>↑ influenza-specific CD8$^+$ T cell responses<br>↑ epithelial regeneration | |
| Scheer 2014 [117] | Influenza A + Pneumonia virus of mice | Not reported | *Sm* | ↓ mortality and weight loss<br>↑ mucus production and goblet cell hyperplasia in lungs<br>↓ lung viral burden | |

†*Sm* = *S. mansoni*; *Sh* = *S. haematobium*; *Sj* = *S. japonicum*; *Hp* = *Heligmosomoides polygyrus*
*Statistically significant

## Human papillomavirus

**Schistosomes and prevalence of HPV.**   Two small studies suggested possible interactions between *S. haematobium* and HPV. In 37 women followed for 5 years, *S. haematobium* was associated with development of high-grade squamous intraepithelial neoplasia but not with high-risk HPV persistence (**Table 9**) [119]. Genital HPV DNA was also associated with reported or confirmed *S. haematobium* in Tanzania, but only 6 women had confirmed schistosome infection [120]. Therefore, despite both pathogens affecting the genital mucosa, interactions between *S. haematobium* and HPV remain poorly understood.

**Schistosomes and HPV vaccination.**   Two studies have investigated the effect of schistosome infection on HPV vaccination. A small number of *S. mansoni*-infected baboons had lower HPV IgG antibody titers than baboons without schistosome infection, though praziquantel treatment prior to vaccination eliminated these differences [121]. However, a large study reported no difference in anti-HPV antibody titers seven and 12 months after HPV vaccination in 298 girls and women aged 10–25 with versus without *S. mansoni* infection [122]. Importantly, diagnosis of schistosomes by egg excretion, which is poorly sensitive in women [123], could have led to misclassification of infected women. Additional studies to understand schistosome-HPV interactions should be prioritized.

## Human T-cell leukemia virus type 1 (HTLV-1)

**Effects of schistosome infection on host immune response to HTLV-1.**   Three studies examined effects of *S. mansoni* coinfection on Th1 proinflammatory responses in HTLV-1, which are associated with development of tropical spastic paralysis [124]. In two studies, the addition of *S. mansoni* antigens to PBMC cultures from HTLV-1-infected adults caused increased IL-10 and decreased IFN-γ and CXCL9, important for Th1 cell recruitment [125,126]. Another reported decreased IFN-γ and increased IL-10 in unstimulated PBMC cultures of adults with HTLV-1-helminth co-infection versus with HTLV-1 alone [127], as well as decreased HTLV-1 proviral load in PBMCs of those with coinfection [124]. These data fit with one study's clinical documentation of schistosome or strongyloides infections seven-fold more frequently in HTLV-1 disease carriers than in HTLV-1 infected persons who developed tropical spastic paralysis [124], suggesting that decreased Th1 and increased T regulatory could prevent the development of inflammatory paralysis. In this instance, HTLV-1 could be another viral infection that is positively impacted by schistosome coinfection and confirmatory studies are needed.

## Rubeola virus (measles)

**Schistosomes and measles vaccination.**   Five studies examined the effect of *S. mansoni* on measles vaccine response. In the first, preschool children with *S. mansoni* infection who received catch-up measles vaccination achieved lower anti-measles IgG titers. Children treated with praziquantel treatment before or at the time of catch-up measles immunization developed higher IgG titers than children treated after immunization, and children who remained schistosome-infected 24 weeks post-vaccination had lower IgG titers than those cured [128]. Lower anti-measles IgG titers were similarly documented in *S. mansoni*-infected schoolchildren [129], though a second study showed robust Th1 and pro-inflammatory post-vaccination cytokine responses to measles antigens [130]. Another study reported that two-year-old children born to mothers with *S. mansoni* had lower anti-measles IgG titers than those born to uninfected mothers [131], although no titer differences were observed in one-year-olds [132] and neither study determined schistosome infection status of the children. In HIV-infected antiretroviral therapy (ART)-naïve adults, anti-measles IgG titers were similar regardless of

**Table 9. Schistosomes and Interactions with HPV, HTLV-1, and Measles.**

| Study | Subject Type | Sample Size | Sex (% Female) | Age in Years (Weighted Mean) | Country | Species† | Differences in Schistosome Infected Compared to Uninfected | Overall Effects of Schisto-some Infection |
|---|---|---|---|---|---|---|---|---|
| **HPV Prevalence** | | | | | | | | |
| Kjetland 2010 [119] | Human | 37 | 100% F | Range, 20–55 | Zimbab-we | Sh | **7 women developed HGSIL and 6 had Sh\*** **No effect on HPV persistence (29% versus 19%)** | Possible increases in HPV prevalence and sequelae; larger studies needed |
| Petry 2003 [120] | Human | 218 | 100% F | Median, 28.4 | Tanzania and Germany | Sh | **Borderline ↑ HPV prevalence (83% in confirmed Sh vs 39% in uninfected)** | |
| **HPV Vaccine** | | | | | | | | |
| Gent 2019 [121] | Baboon | 10 | Not reported | Sub-adult | NA | Sm | **↓ HPV-specific IgG after vaccination in Sh** **IgG restored in baboons given PZQ before vaccine** | Discrepant data; large human study suggests no effect |
| Brown 2014 [122] | Human | 298 | 100% F | Range, 10–25 | Tanzania | Sh, Sm | **No difference in HPV IgG antibody titers** | |
| **HTLV-1** | | | | | | | | |
| Lima 2013 [125] | Human | 26 | 42% F | 48 | Brazil | Sm | **↓ IFN-γ and ↑ IL-10 after schistosome antigen stimulation** | Decreased Th1 cytokines and HTLV-1 viral load in vitro, which may contribute to observed lower rates of tropical spastic paralysis |
| Lima 2017 [126] | Human | 38 | 76% F | 47.2 | Brazil | Sm | **↓ CXCL9 after schistosome antigen stimulation** | |
| Santos 2004 [127] | Human | 120 | Not reported | Not reported | Brazil | Sm | **↓ IFN-γ and ↑ IL-10 in unstimulated cultures** | |
| Porto 2005 [124] | Human | 70 | Approx-imately 15% F | Approx-imately 46 | Brazil | Sm | **↓ IFN-γ and ↑ IL-10** **↓ frequency of CD4+ and CD8\* IFN-γ secreting cells** **↓ HTLV-1 proviral DNA load in PBMCs** | |
| **Response to Rubeola (Measles) Vaccine** | | | | | | | | |
| Tweyon-gyere 2019 [128] | Human | 239 | 53% F | 3.9 years | Uganda | Sm | **↓ anti-measles IgG in Sm at one week; lower IgG trend at 6 months** **↓ % with protective antibody levels in Sm** **↓ anti-measles IgG after PZQ** | Active schisto-some infection during measles vaccine may impair anti-measles antibody response. Maternal Sm infection did not consistently impact children's measles vaccine response. Adults with Sm did not have decreased measles IgG. |
| Nono 2018 [129] | Human | 85 | Not reported for sub-study | Range, 5–18 | Came-roon | Sh, Sm | **↓ anti-measles IgG in Sm but not in Sh (children had been previously vaccinated)** | |
| Jiz 2013 [130] | Human | 104 | Not reported | Range, 7–16 | Phili-ppines | Not listed | **No difference in post-vaccine anti-MMR cytokine responses in schistosome infected** | |
| Ondigo 2018 [131] | Human | 99 | Not reported | 2 years | Kenya | Sm | **↓ anti-measles IgG in 2-year-olds whose mothers had Sm during pregnancy** | |
| Kizito 2013 [132] | Human | 711 | Not reported | 1 year | Uganda | Sm | **No difference in post-vaccine anti-measles IgG in 1-year-olds whose mothers had Sm during pregnancy** | |
| Storey 2017 [133] | Human | 100 | 82% F | 41 | Kenya | Sm | **No difference in anti-measles IgG in HIV-infected adults with Sm** | |

\*HGSIL = High grade squamous intraepithelial lesion

†Sm = S. mansoni; Sh = S. haematobium

schistosome infection status [133]. Overall, these results suggest that active schistosome infection at the time of measles vaccination may impair anti-measles antibody response. Schistosome infection of the mother and schistosome infection acquired in adulthood seem to have limited effects compared to the relatively stronger negative impact of schistosome infection during childhood at the time of measles vaccination.

In summary, the 103 studies included in this review illustrate the breadth of data and the complex impact of schistosomes on host anti-viral immunity. Ramifications on viral acquisition, virulence, and prevention are diverse and vary not only with host factors but also with time-sensitive factors including the chronicity of the schistosome infection and elapsed time since anti-schistosome treatment. In addition, interactions of schistosomes with viruses such as HCV and HIV have been relatively well-studied, while only a few studies each have examined effects on other highly prevalent viruses including HPV, KSHV, and influenza.

## Discussion

Studies included in this review consistently demonstrate that schistosome infection impairs host immune response to some anti-viral vaccines, skews anti-viral immunity, and impacts, possibly both positively and negatively, virulence of viral infections depending on the virus and the site of pathology (**Fig 2**). With regards to viral prevention, active schistosome infection seems to have mostly harmful effects by decreasing cellular and humoral responses to anti-viral vaccinations and increasing risk for acquisition of certain viruses, including HIV and KSHV. Once a virus has been acquired, schistosome coinfection induces Th2-dominant immune responses that may exacerbate severity of some viral infection outcomes, particularly hepatic damage by hepatidities and enteric viruses. However, in other viral infections including HIV, HTLV-1, and respiratory viruses, schistosome infection may mitigate disease severity. Taken together, these findings urge comprehensive consideration of effects of co-infections in endemic settings where viral and parasitic infections frequently converge, and point to the need for further research (**Table 10**). This is especially important given the high prevalence of schistosomiasis and helminth infections globally [134,135].

One of the most important potential public health impacts of our findings is that schistosome infection decreases the cellular and humoral immune responses to several anti-viral immunizations including hepatitis B and measles. Poorer responses to poliovirus, rotavirus, and Ebola vaccines have been widely reported in low-income helminth-endemic countries [136–138]. For example, oral rotavirus vaccine efficacy was 49% in Malawi versus 77% in higher-income countries [139], and the effects of schistosome infection are not known. Additional data are urgently needed to determine whether treatment of schistosome infection prior to vaccine administration could improve efficacy, as suggested by multiple studies in this review, and to optimize treatment timing in order to maximize resolution of immune alterations and elimination of schistosome eggs while limiting risk of schistosome re-infection. Coupling staged anti-schistosome treatment and subsequent vaccination has potential to confer the dual benefit of lowering schistosome-associated pathology and improving viral prevention. Schistosome treatment may improve responses to bacterial vaccines such as tetanus and BCG as well [41,140,141].

Studies have also consistently reported that schistosome-hepatitis coinfection increases severity of liver disease, and limited evidence suggests that anti-schistosome treatment could prevent or alleviate liver fibrosis in schistosome-hepatitis co-infected persons. Therefore, screening and treating for schistosome infection immediately in individuals diagnosed with hepatitis viruses could avert severe liver disease.

While many studies demonstrate potential benefits of anti-schistosome treatment, others invite further research in the context of some viral infections. Our synthesis of articles suggests

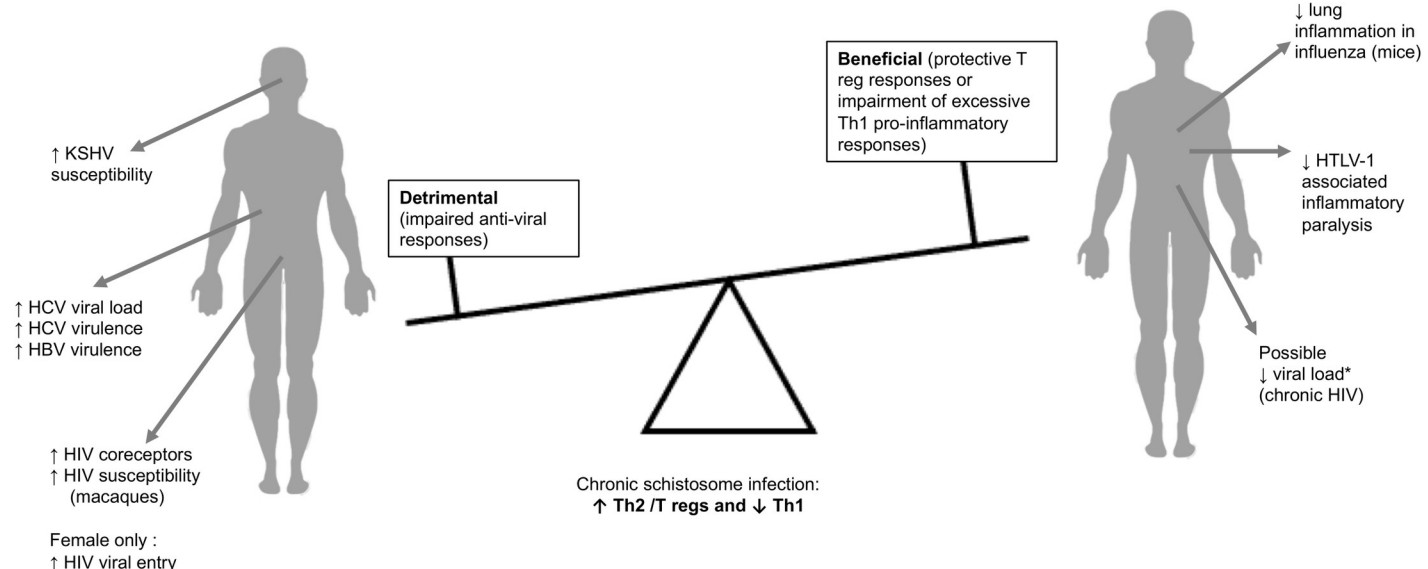

* Indicates that not all studies were statistically significant and more research is needed.

**Fig 2. Chronic schistosome infection was associated with impaired anti-viral responses in some viral infections.** In others, chronic schistosome infection appeared to offer protection against viral infection attributable to T regulatory responses or to control of excessive Th1 inflammatory responses.

that treatment of schistosome infection could increase HIV viral load in chronic HIV, and that coinfection with schistosomes could reduce mortality in HIV-infected individuals. Both findings are under-studied and lack statistical significance; further research is imperative to fully understand their implications. Their existence, however, does suggest that mechanistic studies on the potential beneficial effect of schistosome coinfection in chronic HIV are needed to pave the way for novel immunoregulatory interventions.

Across an array of viruses and mammals, articles in this review consistently report Th1-dominant immune responses to viral mono-infection, versus Th2 and T regulatory-dominant responses with schistosome-virus coinfection. These conserved immune responses to virus-parasite infections may suggest strategies that could harness the beneficial effects of parasites in some viral infections to improve clinical outcomes. The strong T regulatory component of chronic schistosome infection may be particularly important, and studies in this review have associated a T regulatory environment with decreased hepatic damage from HBV [36] and an

**Table 10. Broad Priority Research Areas for Schistosome-Virus Coinfections.**

| Topic | Priority Research Needs |
|---|---|
| Vaccines | Vaccine efficacy in schistosome-endemic settings before schistosome infection is acquired, or in LMIC settings in which schistosomes are endemic versus LMIC settings in which schistosomes are not endemic<br>Effect of schistosome infection on other anti-viral vaccines, including rotavirus and poliovirus<br>Optimal timing for praziquantel treatment prior to vaccination |
| Schistosome-Virus Immunology | Precise timing of the dynamics of the human immune response to schistosome infection, and to schistosome-virus coinfection<br>Roles of mucosal and systemic immune cell populations in mediating response to virus in schistosome-infected populations |
| Schistosome-Virus Treatment | Antiviral effects of schistosome-induced host immunoregulatory molecules or cells<br>Antiviral effects of immunomodulatory parasite effector molecules |

HTLV-1 carrier rather than pro-inflammatory disease state [124]. T regs also have decreased HIV-1 R5 virus susceptibility [142], which could limit cell-cell spread and slow HIV disease progression. Other immune cell populations upregulated in chronic schistosome infection, such as alternatively activated macrophages [109], and immunomodulatory parasite effector molecules that incite Th2 and T regulatory responses (the subject of a recent comprehensive review) [143], may have promising anti-viral therapeutic efficacy. Efforts to identify and test immune cell products and schistosome immunomodulatory secretions should be included in ongoing global research to prevent and treat viral infections. In addition, recent developments in the area of controlled human schistosome challenge model [19] are expected to facilitate more precise mapping of the host response throughout acute and chronic schistosome infection. Although these controlled schistosome infections use only male worms and therefore may induce fewer Th2 and T reg responses because schistosome eggs are not produced, they nonetheless provide a potent resource to understand immune responses to coinfections.

Effects of timing, duration, and dose of both schistosome infection and viral infections could also influence the manifestation of schistosome-viral coinfection. Individuals may be less likely to become infected with schistosome infection before acquiring viral infections that are common in childhood, such as rotavirus. Conversely, they may be more likely to become infected with schistosome infection during childhood before acquiring sexually transmitted infections, such as HIV and HPV. Amount of time with untreated schistosome infection may also impact the host response to coinfection. Thus, incorporating age prevalence curves of viral infections in schistosome-endemic regions may be useful when considering public health interventions to address coinfection.

This review has limitations. Only studies that documented active schistosome infection by microscopy or antigen testing, but not by antibodies, were included due to antibodies remaining positive after clearance of infection. This may have caused us to underestimate effects of schistosome infection, because individuals who had cleared schistosome infections that previously affected their viral control would have been considered schistosome-uninfected. It also could preclude capturing long-term immunological effects of past infections. Effects of environmental, sociodemographic, and economic factors associated with schistosome infection also could not be evaluated [144]. Further, only studies examining the impact of schistosome infection on viral infections were included. Those assessing the impact of viruses on schistosome infection were beyond the scope of this review, as were those not published in English. Finally, publication bias could have led to the non-publication of studies that showed no impact of schistosomes on viral infections and thereby would not have been included.

In summary, this review highlights the major and complex effects of schistosome infection on host response to viruses, encompassing viral acquisition, immune response, virulence, and prevention. Considering the impact of parasitic infections on viral control could have important public health implications for anti-viral vaccination strategies and control of viral infections globally.

## Supporting information

**S1 Table. List of Excluded Studies.**
(DOCX)

**S2 Table. Characteristics, Key Findings, and Quality Assessment of Included Studies.**
(DOCX)

**S1 Text. OVID MEDLINE ALL Complete Search Strategy and Grey Literature Search Strategy.**
(DOCX)

**S2 Text. Decision Rules for Data Selection and Extraction Processes.**
(DOCX)

## Author Contributions

**Conceptualization:** Katherine Klemperer, Jennifer A. Downs.

**Data curation:** Keith Mages, Andrea Chalem.

**Formal analysis:** Brooke W. Bullington, Katherine Klemperer, Humphrey D. Mazigo, John Changalucha, Saidi Kapiga, Peter F. Wright, Maria M. Yazdanbakhsh, Jennifer A. Downs.

**Funding acquisition:** Jennifer A. Downs.

**Investigation:** Brooke W. Bullington, Jennifer A. Downs.

**Methodology:** Katherine Klemperer, Keith Mages, Jennifer A. Downs.

**Software:** Keith Mages.

**Supervision:** Jennifer A. Downs.

**Visualization:** Brooke W. Bullington, Andrea Chalem.

**Writing – original draft:** Brooke W. Bullington, Jennifer A. Downs.

**Writing – review & editing:** Keith Mages, Andrea Chalem, Humphrey D. Mazigo, John Changalucha, Saidi Kapiga, Peter F. Wright, Maria M. Yazdanbakhsh.

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
