## [Decision Letter · Decision Letter 0]

25 Mar 2021

Dear Ms. Bullington,

Thank you very much for submitting your manuscript "Effects of Schistosomes on Host Anti-Viral Immune Response and the Acquisition, Virulence, and Prevention of Viral Infections: a Systematic Review" for consideration at PLOS Pathogens. As with all papers reviewed by the journal, your manuscript was reviewed by members of the editorial board and by several independent reviewers. The reviewers appreciated the attention to an important topic. Based on the reviews, we are likely to accept this manuscript for publication, providing that you modify the manuscript according to the review recommendations.

Sincerely,

Michael H. Hsieh

Guest Editor

PLOS Pathogens

P'ng Loke

Section Editor

PLOS Pathogens

Kasturi Haldar

Editor-in-Chief

PLOS Pathogens

orcid.org/0000-0001-5065-158X

Michael Malim

Editor-in-Chief

PLOS Pathogens

orcid.org/0000-0002-7699-2064

Reviewer Comments (if any, and for reference):

Reviewer's Responses to Questions

**Part I - Summary**

Reviewer #1: This manuscript reviews published (and some unpublished) research on the effects of schistosome infections on viral diseases and anti-viral immunizations. It does a good job of bringing a couple of decades worth of publications together in one location. Unfortunately, because of a lack of similar methods across research studies, no meta analyses are possible, making it difficult to draw definitive conclusions. The manuscript is overall well written and informative but there are a number of smallish issues that should be addressed.

Reviewer #2: This is a scoping systemic review of a very dense subject that broadly synthesizes the impact of schistosomiasis on immune responses to highly prevalent viral infections as well as responses to vaccines for some of these. This was an incredible effort with an initial look at thousands of papers for conclusion and will be a significant contribution to the extant literature on this expansive and often discordant subject of the effects of schistosomiasis on viral infection virulence, response to vaccines, response to therapeutics, and risk of infection (HIV). The authors nicely provide the clinical or public health significance of the reviewed studies, such that there are also translational conclusions that can be drawn, in particular, when treating schistosomiasis with praziquantel can improve the clinical course and response to vaccines for specific viruses prevalent in regions where schistosomiasis is co-endemic. Throughout this manuscript, the authors do a very nice job of highlighting differences between human and animal studies and implications, as well as differences in expected immune response in humans during acute vs. chronic schistosomiasis infection. They also do not overstate findings of the review and highlight where evidence to support a particular finding is strong or less strong and in the latter case, provide insightful and evidence based reasons for discrepancy in studies. Finally the tables are very helpful, synthesizing design and results of studies and providing reference.

**Part II – Major Issues: Key Experiments Required for Acceptance**

Reviewer #1: As this was a review, there are no experiments and as mentioned above, the broad nature of the topic precludes any secondary analyses. The only issue approaching major status is the need to proceed with greater caution on suggestions that schistosomiasis coinfection may be beneficial with respect to some viral infections as this may lead to excuses not to execute the WHO 2030 roadmap that expands praziquantel access and distribution. In a few situations (e.g., HTLV), the immunomodulatory effects of chronic schistosomiasis may decrease the pathology caused by the viral infection. However, it is not ethically or logistically tenable that treatment should be withheld from certain individuals. A more nuanced way to approach this possibility would be to encourage research into the factors that are beneficial (e.g., what aspect of schistosome infections increase Treg cells) and employ these separately from any possibility that treatment would be denied to some individuals. Specifically, decrease in HIV viral load should be removed from the "beneficial" side of Figure 2. The data that the authors present in the paper from references 97 and 103 is far from robust--it is not even statistically significant. A much stronger body of literature would be needed to justify inclusion on that side of the balance. It would be fine however to note that the data are equivocal--perhaps moving the viral load designation over the fulcrum is more appropriate. Again, I do not think the authors would want a control program using this figure to justify failure to distribute praziquantel.

Reviewer #2: Authors should state in methods (or discussion) whether their definition of virulence (“severity of clinical disease in co-infected subjects”) coincided well with definitions used in selected/included primary manuscripts.

Page 7 – it is stated that Abruzzi et. al.” concluded that numerous studies in patients

with schistosome-HCV coinfection indicate that schistosome-hepatitis coinfections have

multiplicative, rather than additive, effects on liver disease.”

Suggest being sure that appropriate statistical approaches were used to determine “multiplicity” (or synergistic effects even if worsening”) rather than purely additive.

The Hepatitis B and C section reviews the role of schistosomiasis in both viral virulence or severity of infection as well as other key clinical outcomes such as vaccine response to Hepatitis B and highlights papers that provide an “actionable” approach to treatment with praziquantel at the time of vaccination.

The section on HIV nicely reviews the immunologic mechanistic support, both mucosal and systemic, for the observation that schistosomiasis infected women are at increased risk for HIV acquisition. This is a very dense literature broadly and it is nicely synthesized here. It too provides a “translational” framework whereby treating schistosomiasis could both decrease risk of acquisition and also modulate immune responses to HIV infection and vaccine candidates. Table 3 also nicely splits the studies based on human mucosal and systemic immune effects and animal models. Throughout they do a very nice job of highlighting differences between human and animal studies and implications, as well as differences in expected immune response in humans during acute vs. chronic schistosomiasis infection.

The HIV acquisition section also nicely summarizes limitations of studies and why there have been many conflicting reports including lack of control for confounders and species studied (haematobium vs. mansoni).

Page 16

Authors provide a solid hypothesis with respect to discordant findings of studies addressing role of schistosomiasis in HIV progression and CD4 counts that references timing of co-infections and intensity.

The figure provides a helpful overview of the complex immunological mechanisms.

Table 9 sets priorities for future research and summarizes where impact of treatment for schistosomiasis may impact key aspects of viral vaccinology and virulence.

**Part III – Minor Issues: Editorial and Data Presentation Modifications**

Reviewer #1: There are a number of minor issues to be addressed. Mainly they are related to being clear about which comparisons are being made. For example, towards the end of page 7, the authors state that people with HBV-S. mansoni coinfections who receive praziquantel treatment more frequently cleared serum HbSAg than those with HBV (infection) alone. It seems like the more appropriate comparison would be in the coninfected individuals before and after treatment rather than between treated individuals and those who never had schistosomiasis to begin with. Please clarify.

Similarly, the last sentence of the paragraph at the top of page 9 suggests that treatment with praziquantel could improve vaccine responsiveness and decrease global HBV burden but this is only true in people that had previously had schistosomiasis. Certainly this is implied and the authors probably wanted to avoid being pedantic but as a standalone sentence it makes no sense and the clause "in persons with schistosomiasis" or "in schistosomiasis endemic areas" should be added to the end of the sentence.

Another example is the last sentence on page 18: "scenario in which schistosome-viral coinfection may be surprisingly beneficial". It would be better to have neither infection. What is surprisingly beneficial is a schistosome coinfection compared to viral infection alone.

Second to last sentence in first paragraph on page 19: "...but only 6 women had confirmed infection." With what? HPV or schistosomes? Second sentence of next paragraph "...than uninfected baboons." Better to say than baboons that only had HPV infection.

Third sentence in last paragraph on page 19--please indicate the timing of the praziquantel treatment that restored IgG titers. Were this titers actually "restored" in a given individual or did the praziquantel treatment group simply have a higher level of IgG than the untreated schistosome-infected group?

First line of page 23: presumably the intent is that treatment of schistosomiasis alleviates liver fibrosis in persons with hepatitis. It is no surprise that treating schistosomiasis prevents schistosomiasis-related fibrosis. Just be clear.

In discussing the human schistosome challenge model on page 23, it is important to note that these are not patent infections, thus some of the presumed benefit of the Th2 and T reg responses associated with egg deposition may not in fact be available.

Please read back through the whole manuscript and correct these and other unclear comparisons. This can be annoying but it is necessary to be very clear about which infection is being referred to in coinfection studies.

Other small errors needing correction:

--the first two entries in table 1 include the journal, the rest don't

--a quick glance shows that several references (21, 110, 130) are incomplete and other citations (112, 128) are not valid references. Please go back through and check the entire bibliography.

--in figure 2, there seems to be words missing at the end of the "Beneficial" box. Also, the increase in HIV coreceptors and macaque susceptibility is not limited to females alone. What about transmission in this figure? An older paper by Leuschter showed increased CD4 cells in semen of men with high intensity S. haematobium infection and citation 85 shows greater risk of transmitting HIV to HIV-negative partners by persons with schistosome infection.

--Table 9 first sentence on vaccine efficacy seems to have an addition or deletion word mutation (are versus). THe point about precise timing in the next block brings up an important point that probably merits a paragraph in the discussion. Age prevalence curves of the different infections may be key. For example, a person is likely to get exposed to rotavirus and childhood vaccinations prior to becoming infected with schistosomes but conversely is more likely to acquire a schistosome infection in childhood before they are exposed to sexually transmitted viruses like HPV, HCV, and HIV. Exposure to respiratory viruses may be more concurrent. Both the timing and the dose of the various infections could influence the manifestation of the coinfection.

Reviewer #2: Page 5 – for reference to HCV specific intrahepatic TH1 responses, should mention these were human subjects with biopsy if that how the this was ascertained.

In figure one would change “piece” as in “review piece” to article.

PLOS authors have the option to publish the peer review history of their article (what does this mean?). If published, this will include your full peer review and any attached files.

Reviewer #1: No

Reviewer #2: No

Figure Files:

Data Requirements:

Reproducibility:

References:

---

## [Editor Report · Decision Letter 1]

13 Apr 2021

Dear Ms. Bullington,

We are pleased to inform you that your manuscript 'Effects of Schistosomes on Host Anti-Viral Immune Response and the Acquisition, Virulence, and Prevention of Viral Infections: a Systematic Review' has been provisionally accepted for publication in PLOS Pathogens.

Best regards,

Michael H. Hsieh

Guest Editor

PLOS Pathogens

P'ng Loke

Section Editor

PLOS Pathogens

Kasturi Haldar

Editor-in-Chief

PLOS Pathogens

orcid.org/0000-0001-5065-158X

Michael Malim

Editor-in-Chief

PLOS Pathogens

orcid.org/0000-0002-7699-2064
---

## [Editor Report · Acceptance letter]

14 May 2021

Dear Ms. Bullington,

We are delighted to inform you that your manuscript, "Effects of Schistosomes on Host Anti-Viral Immune Response and the Acquisition, Virulence, and Prevention of Viral Infections: a Systematic Review," has been formally accepted for publication in PLOS Pathogens.

Best regards,

Kasturi Haldar

Editor-in-Chief

PLOS Pathogens

orcid.org/0000-0001-5065-158X

Michael Malim

Editor-in-Chief

PLOS Pathogens

orcid.org/0000-0002-7699-2064